# No-reference quality assessment for image-based assessment of economically important tropical woods

Heshalini Rajagopal[1], Norrima Mokhtar[1]*, Tengku Faiz Tengku Mohmed Noor Izam[1], Wan Khairunizam Wan Ahmad[2]

1 Department of Electrical Engineering, Faculty of Engineering, University of Malaya, Kuala Lumpur, Malaysia, 2 School of Mechatronic Engineering, Universiti Malaysia Perlis, Arau, Malaysia

⊙ These authors contributed equally to this work.
* norrimamokhtar@um.edu.my

**Data Availability Statement:** The images and data used in this study are publicly available in https://github.com/Heshalini/Wood-Image-Quality/tree/b03346da9d1efb887ec6d390f235822cc9470f49.

## Abstract

Image Quality Assessment (IQA) is essential for the accuracy of systems for automatic recognition of tree species for wood samples. In this study, a No-Reference IQA (NR-IQA), wood NR-IQA (WNR-IQA) metric was proposed to assess the quality of wood images. Support Vector Regression (SVR) was trained using Generalized Gaussian Distribution (GGD) and Asymmetric Generalized Gaussian Distribution (AGGD) features, which were measured for wood images. Meanwhile, the Mean Opinion Score (MOS) was obtained from the subjective evaluation. This was followed by a comparison between the proposed IQA metric, WNR-IQA, and three established NR-IQA metrics, namely Blind/Referenceless Image Spatial Quality Evaluator (BRISQUE), deepIQA, Deep Bilinear Convolutional Neural Networks (DB-CNN), and five Full Reference-IQA (FR-IQA) metrics known as MSSIM, SSIM, FSIM, IWSSIM, and GMSD. The proposed WNR-IQA metric, BRISQUE, deepIQA, DB-CNN, and FR-IQAs were then compared with MOS values to evaluate the performance of the automatic IQA metrics. As a result, the WNR-IQA metric exhibited a higher performance compared to BRISQUE, deepIQA, DB-CNN, and FR-IQA metrics. Highest quality images may not be routinely available due to logistic factors, such as dust, poor illumination, and hot environment present in the timber industry. Moreover, motion blur could occur due to the relative motion between the camera and the wood slice. Therefore, the advantage of WNR-IQA could be seen from its independency from a "perfect" reference image for the image quality evaluation.

## Introduction

Wood is a plant tissue consisting of a porous and fibrous structure. It is widely used as a source of energy, and for furniture making, millwork, flooring, building construction, and paper production [1]. Thousands of wood producing tree species are present, which comprise materials of distinct physical characteristics in terms of structure, density, colour, and texture [2]. These

**Funding:** The author(s) received no specific funding for this work.

characteristics define the preferred usages and monetary values of the trees [3]. Furthermore, although the timber production at high latitudes is based on a small number of species, a wide range of tropical forests is present. For example, conifers of the genus Pine are widespread in the Northern Hemisphere. Subsequently, this phenomenon leads to the production of moderate-priced wood of high resin content, which is widely used for the making of indoor furniture.

Discovered in the native of Central America, Bocote (*Cordia gerascanthus*) is used to produce high-cost hardwood, which is suitable for high-quality furniture and cabinetry due to the glossy finish created by the oily surface of the wood. Meanwhile, the rosewood (*Dalbergia* sp.) is another high-cost wood, which is sought for instrument making and flooring due to its high strength and density. Provided that each wood species consists of various price and characteristics, misclassification of the wood could lead to financial losses. Therefore, the correct identification of the different wood species is essential.

Although the recognition of wood species is traditionally performed by humans [4], the process of it is time-consuming and incurs a high cost to the lumber industry. Therefore, various algorithms have been developed for automatic recognition of wood samples [1, 2, 5, 6]. A scope is present for the improvement in the accuracy of automatic wood recognition systems through high-quality microscopy images, which are sometimes pre-processed to enhance the recognition. However, the processes of image enhancement require more time and may impart a checkerboard artefact to the wood images [7]. Besides, the environment of timber factories is surrounded by dust, poor illumination, and heat [8], which lead to the degradation of the image quality. Therefore, a suitable Image Quality Assessment (IQA) metric is essential to evaluate the captured images before proceeding to the pipeline for recognition algorithms.

Image Quality Assessment (IQA) may be specified into two categories, namely subjective and objective evaluations. Subjective evaluation occurs when the images are evaluated by human, who provide scores based on their perception on the image quality, while objective evaluation involves mathematical algorithms to calculate the quality score for the images [9]. Although subjective evaluation is regarded as the gold standard in IQA, it is not practical in the industrial setting due to the high cost and long duration required. Therefore, an incentive is made to develop objective evaluation procedures of the comparable quality to subjective IQA evaluation [9].

The objective evaluation consists of three categories, namely Full-Reference-IQA (FR-IQA), Reduced Reference-IQA (RR-IQA), and No-Reference/Blind-IQA (NR-IQA) [10, 11]. Specifically, FR-IQA evaluates an image by comparing the image with its reference image, while NR-IQA evaluates an image without involving reference images. Meanwhile, RR-IQA assesses an image using partial information from reference images [12]. Notably, NR-IQA is the most suitable metric used to assess wood images due to the impediments (dusty environment and poor illumination) to the achievement of high-quality images in the environment of lumber mills.

Several NR-IQA metrics were proposed, such as Blind/Referenceless Image Spatial Quality Evaluator (BRISQUE) [13], deepIQA [12] and Deep Bilinear Convolutional Neural Networks (DB-CNN) [14]. Specifically, BRISQUE [13] is trained with Generalised Gaussian Distribution (GGD) and Asymmetric Generalised Gaussian Distribution (AGGD) features by Support Vector Regression (SVR) model for modelling of the images in the spatial domain. Furthermore, deepIQA and DB-CNN are CNN-based NR-IQAs, in which the deepIQA is trained end-to-end. It also involves 10 convolutional layers, five pooling layers for feature extraction, and two fully connected layers for regression [12]. Meanwhile, DB-CNN is trained by two sets of features, namely CNN for synthetic distortions (S-CNN) and VGG-16, which are bi-linearly pooled to measure the quality of the image [14]. However, provided that a limited number of

labelled training data often leads to overfitting problem in CNN, the CNN-based NR-IQA model requires a larger size of the training database [14].

Accordingly, an investigation was conducted on the NR-IQA procedure, which was based on a widely-used NR-IQA, the Blind/Referenceless Image Spatial Quality Evaluator (BRIS-QUE) model. As an IQA model, BRISQUE was not a distortion-specific model. Instead, it considered the luminance and image features of the natural images [13]. Furthermore, the BRISQUE model was trained with subjective scores to enable the emulation of human judgement on the quality of the images. Provided that BRISQUE was trained to evaluate natural images, it was not optimal for the assessment of wood images. Therefore, an NR-IQA was proposed specifically for the assessment of wood images. Following that, the proposed metric, Wood NR-IQA (WNR-IQA) was then compared with BRISQUE [13], deepIQA [12], DB-CNN [14], and five types of established FR-IQA metrics, such as Structural Similarity Index (SSIM) [15], Multiscale SSIM (MS-SSIM) [15], Feature Similarity (FSIM) [16], Information Weighted SSIM (IW-SSIM) [17], and Gradient Magnitude Similarity Deviation (GMSD) [18]. The relative performances of the WNR-IQA, BRISQUE, deepIQA, DB-CNN, and FR-IQAs were identified based on the correlation between the human mean opinion scores (MOS) and the metrics. In this case, the Pearson Linear Correlation Coefficient (PLCC) [19] and Root Mean Squared Error (RMSE) [20] were used.

## Materials and methods

### Training and testing database

To cater to image quality assessment, specifically for wood images, Generalized Gaussian Distribution (GGD) and Asymmetric Generalized Gaussian Distribution (AGGD) features were calculated for wood images. This process involved the subjective MOS obtained from a subjective evaluation for wood images. These GGD, AGGD features, and MOS were used as the training and testing database for the SVR model.

**Wood images.** Ten wood images of ten wood species in the lumber industry, namely *Turraeanthus africanus* (Avodire), *Ochroma pyramidale* (Balsa), *Cordia spp*. (Bocote), *Juglans cinerea* (Butternut), *Tilia Americana* (Basswood), *Vouacapoua americana* (Brownheart), *Cornus florida* (Dogwood), *Cordia spp*. (Laurel Blanco), *Swartzia Cubensis* (Katalox), and *Dipterocarpus spp* (Keruing), were obtained from a public wood database: https://www.wood-database.com/ [21]. The ten wood images are presented in Fig 1.

The images were then converted to grayscale, followed by normalisation of the pixel values to the range of 0–255 to facilitate the application of the same levels of distortion across all the reference images. Furthermore, the images consisted of a matrix of 600 x 600 pixels, which corresponded of an image area of 9525 cm$^2$. The ten reference wood images were then distorted by Gaussian white noise and motion blur to represent the image distortions, which were encountered in the industrial setting. To be specific, Gaussian white noise often arises during the acquisition of wood images due to the sensor noise [22] caused by poor illumination and high ambient temperature in the lumber mill [8]. Meanwhile, wood images were subjected to motion blur upon the presence of relative motion between the camera and the wood slice [6]. Provided that these distortions resulted in a low quality of the wood image, the features of the pores on the wood texture could not be distinguished from one another. As a result, misclassification of the wood species occurred as the feature extractor would not be able to effectively extract distinctive features from the wood texture images [23].

The Gaussian white noise with a standard deviation of $\sigma_{GN}$ and a motion blur with a standard deviation of $\sigma_{MB}$ were applied to the reference images at five levels of distortion. For example, the $\sigma_{GN}$ for Gaussian white noise amounted to 10, 20, 30, 40, 50 and, while the $\sigma_{MB}$

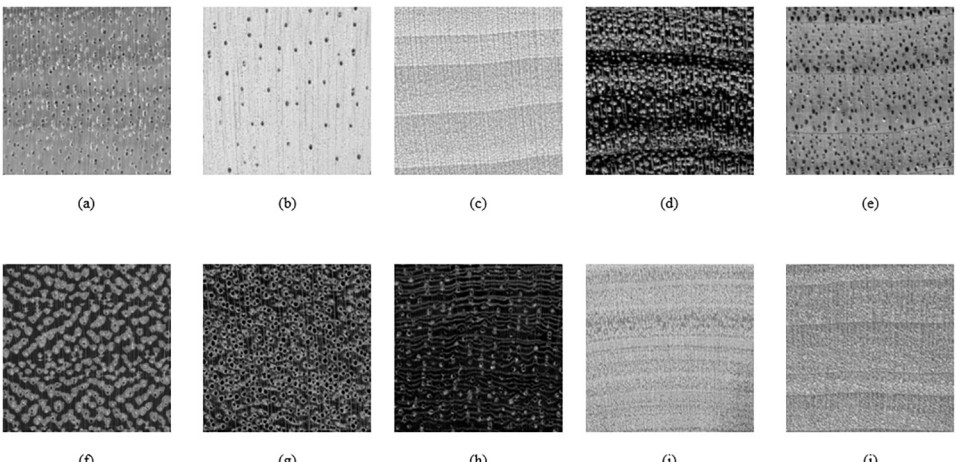

**Fig 1. Ten reference wood images (a)** *Turraeanthus africanus,* **(b)** *Ochroma pyramidale,* **(c)** *Tilia americana,* **(d)** *Cordia spp.,* **(e)** *Juglans cinerea,* **(f)** *Vouacapoua americana,* **(g)** *Dipterocarpus spp.,* **(h)** *Swartzia Cubensis,* **(i)** *Cordia spp.,* **(j)** *Cornus florida.* Reprinted from [21] under a CC BY license, with permission from Eric Meier, original copyright [2007].

for motion blur amounted to 2, 4, 6, 8 and 10. As a result, 110 wood images, ten reference images, 50 images distorted by Gaussian white noise, and 50 images distorted by motion blur were produced. This was followed by the measurement of GGD and AGGD features for these images, which were then used to train the SVR.

**The features of GGD and AGGD.** The Mean Subtracted Contrast Normalized (MSCN), $\hat{I}(m, n)$ was calculated using (1) [12]:

$$\hat{I}(m, n) = \frac{I(m, n) - \mu(m, n)}{\sigma(m, n) + 1} \tag{1}$$

Where, $I(m,n)$ denotes an image, while $\mu(m,n)$ denotes the local mean of $I(m,n)$. While $\sigma(m, n)$ refers to the local variance of $I(m,n)$, $m \in 1, 2, \ldots, M$, $n \in 1, 2, \ldots, N$ refers to the spatial indices, while $M$ and $N$ represent the height and width of image, $I(m,n)$, respectively.

The local mean, $\mu(m,n)$, and local variance, $\sigma(m,n)$, were calculated using the equations in (2) and (3), respectively [12]:

$$\mu(m, n) = \sum_{k=-K}^{K} \sum_{l=-L}^{L} w_{k,l} I_{k,l}(m, n) \tag{2}$$

$$\sigma(m, n) = \sqrt{\sum_{k=-K}^{K} \sum_{l=-L}^{L} w_{k,l} (I_{k,l}(m, n) - \mu(m, n))^2} \tag{3}$$

Where, $w = \{w_{k,l} | k = -K, \ldots, K, l = -L, \ldots, L\}$ denotes a 2-dimension (2D) circularly-symmetric Gaussian weighting function, which was sampled in three standard deviations. This function was then rescaled to unit volume, in which $K$ and $L$ represent the window sizes. It could be seen in Fig 2 that the MSCN, local mean, and local variance of the wood images illustrate the effects of the contrast normalisation.

Furthermore, it is also indicated from Fig 2d that the local variance field, σ, only highlighted the boundary of the pores, while Fig 2e indicates that the MSCN coefficients focused on the key elements of the wood images, including pores and grains, with few low-energy residual object boundaries. According to Mittal et al., the characteristics of MSCN coefficients vary with the occurrence of the distortions [12]. Therefore, the MSCN coefficients were plotted for

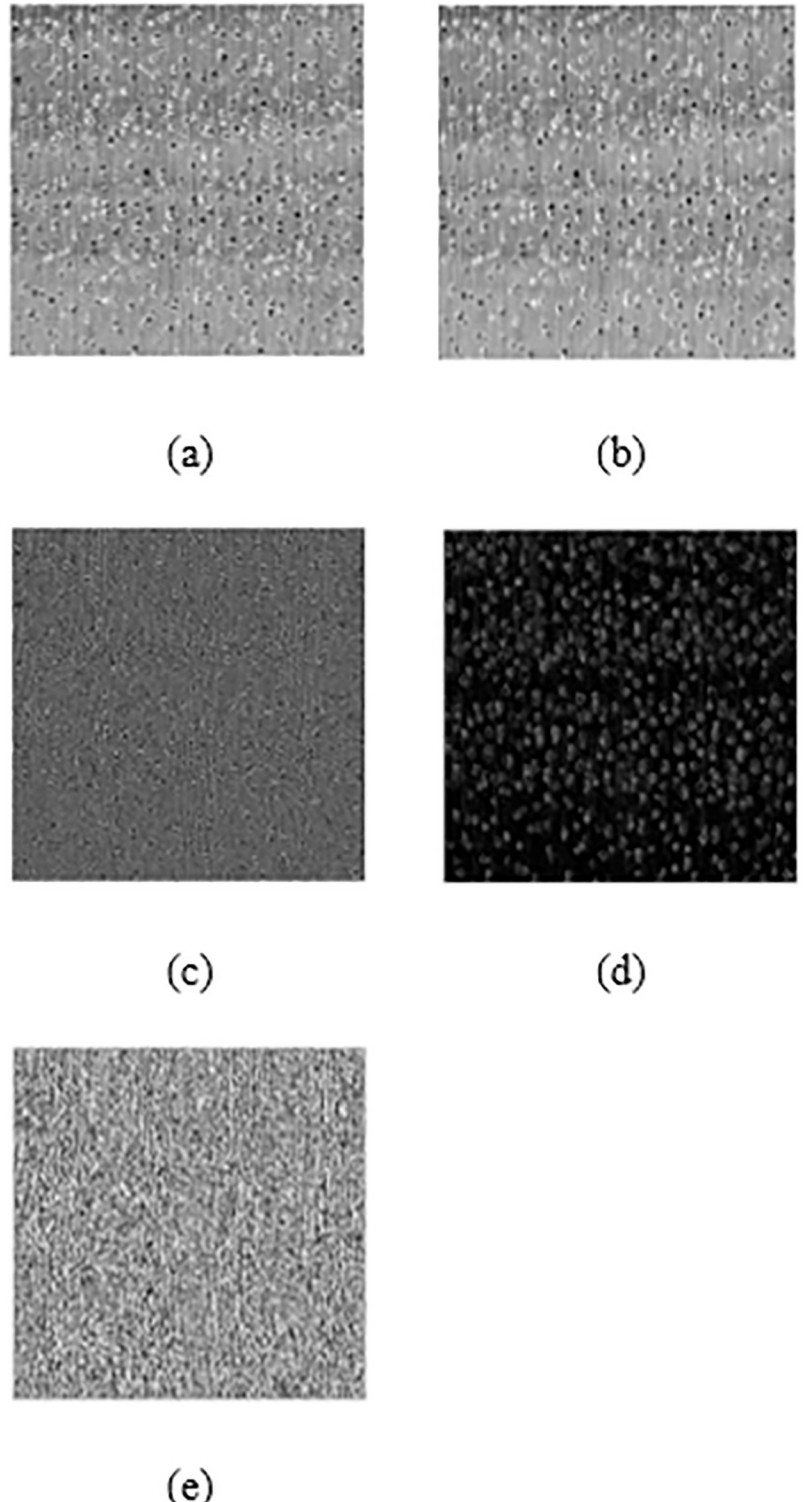

**Fig 2. The effects of the image normalisation procedure on wood image.** Results are focused on the representative case of to Swartzia Cubensis: (a) Original image I, (b) Local mean-field, μ, (c) I − μ, (d) Local variance field, σ and (e) MSCN coefficients. Reprinted from [21] under a CC BY license, with permission from Eric Meier, original copyright [2007].

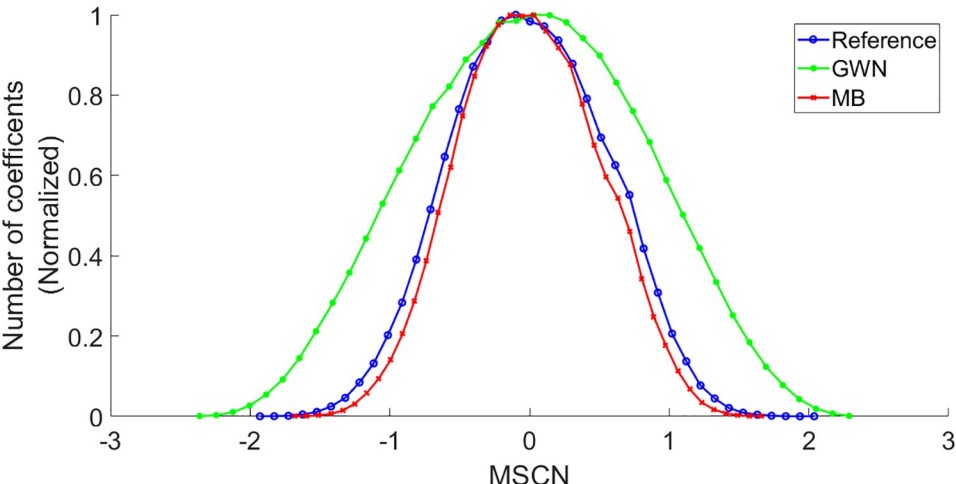

**Fig 3. Histogram of MSCN coefficients for the reference image and distorted images with Gaussian white noise (GWN) and motion blur (MB).** Reprinted from [21] under a CC BY license, with permission from Eric Meier, original copyright [2007].

the reference image, including the images distorted with Gaussian white noise and motion blur to illustrate the resultant changes in the coefficients, as shown in Fig 3.

Based on Fig 3, a Gaussian distribution is presented in the reference images, while the distribution of the images distorted with Gaussian white noise and motion blur consisted of different tail behaviours.

Two types of Gaussian distribution functions were incorporated in this study to accommodate the diverse characteristics of MSCN coefficient, namely the Generalized Gaussian Distribution (GGD) and Asymmetric Generalized Gaussian Distribution (AGGD) [12].

There are two parameters computed for the GGD, where $\alpha$ represents the shape of the distribution and $\sigma^2$ represents the variance. These two parameters are calculated for wood images using the moment-matching principle. The GGD is computed using (4) [24]:

$$f(x; \alpha, \sigma^2) = \frac{\alpha}{2\beta\Gamma\left(\frac{1}{\alpha}\right)} \exp\left(-\left(\frac{|x|}{\beta}\right)^{\alpha}\right) \tag{4}$$

Where

$$x = \hat{I}(m, n) \tag{5}$$

$$\beta = \sigma\sqrt{\frac{\Gamma\left(\frac{1}{\alpha}\right)}{\Gamma\left(\frac{3}{\alpha}\right)}} \tag{6}$$

$$\Gamma(a) = \int_0^\infty t^{a-1} e^{-t} dt, a > 0 \tag{7}$$

Next, the MSCN coefficients were computed throughout the eight orientations, namely horizontal ($H_1$ and $H_2$), vertical ($V_1$ and $V_2$), and diagonal ($D_1$, $D_2$, $D_3$ and $D_4$) as shown in Fig 4.

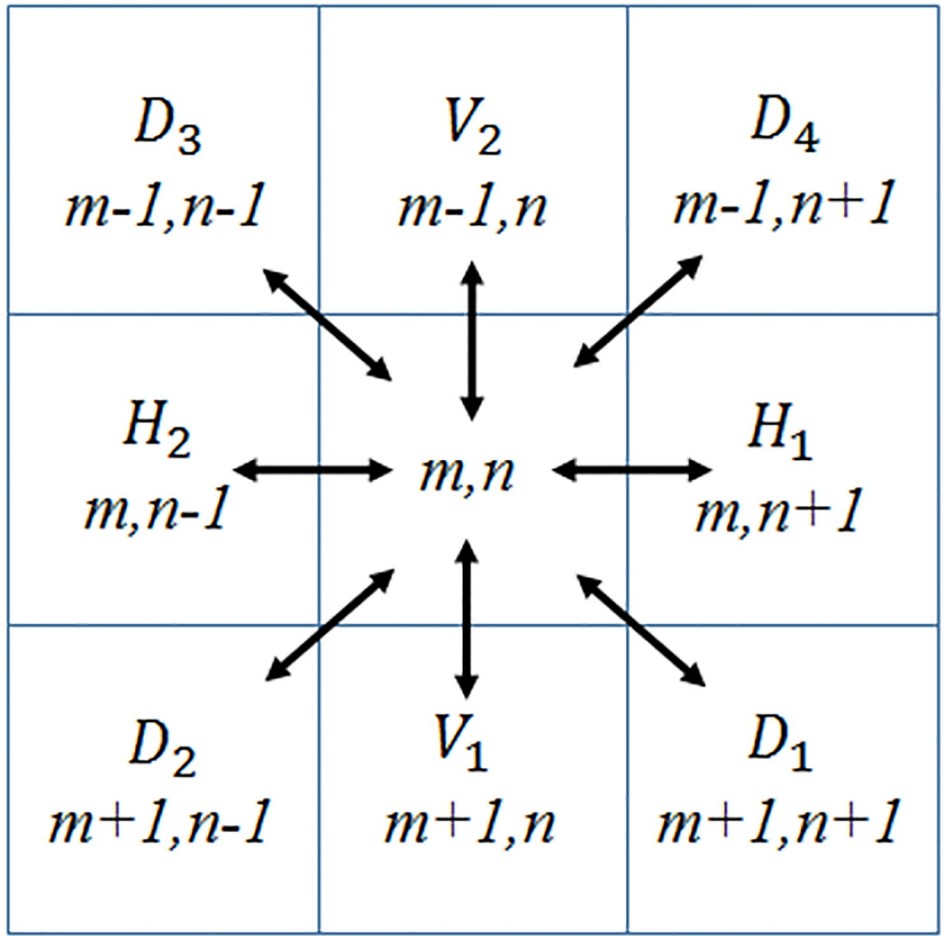

**Fig 4. Eight orientations of neighbourhood pixels of wood images: Horizontal ($H_1$ and $H_2$), vertical ($V_1$ and $V_2$) and diagonal ($D_1$, $D_2$, $D_3$ and $D_4$).** Reprinted from [21] under a CC BY license, with permission from Eric Meier, original copyright [2007].

The computation of the pairwise products of MSCN coefficients throughout the eight orientations: $H_1$, $H_2$, $V_1$, $V_2$, $D_1$, $D_2$, $D_3$ and $D_4$ are shown from Eqs (8) to (15) [12]:

$$H_1(m, n) = \hat{I}(m, n)\hat{I}(m, n + 1) \tag{8}$$

$$H_2(m, n) = \hat{I}(m, n)\hat{I}(m, n - 1) \tag{9}$$

$$V_1(m, n) = \hat{I}(m, n)\hat{I}(m + 1, n) \tag{10}$$

$$V_2(m, n) = \hat{I}(m, n)\hat{I}(m - 1, n) \tag{11}$$

$$D_1(m, n) = \hat{I}(m, n)\hat{I}(m + 1, n + 1) \tag{12}$$

$$D_2(m, n) = \hat{I}(m, n)\hat{I}(m + 1, n - 1) \tag{13}$$

$$D_3(m, n) = \hat{I}(m, n)\hat{I}(m - 1, n - 1) \tag{14}$$

$$D_4(m, n) = \hat{I}(m, n)\hat{I}(m - 1, n + 1) \tag{15}$$

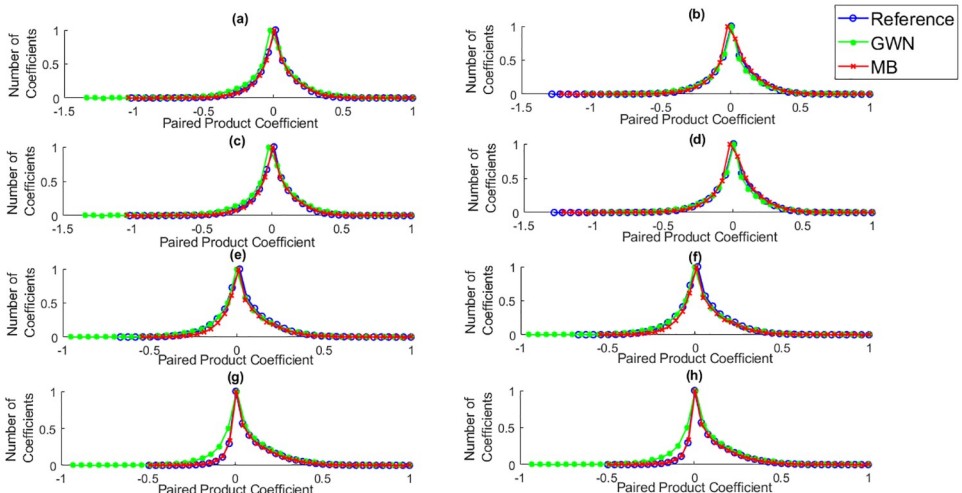

**Fig 5. Histogram of pairwise products of MSCN coefficients in eight directions: (a) $D_1$ (b) $D_2$ (c) $D_3$ (d) $D_4$ (e) $H_1$ (f) $H_2$ (g) $V_1$ (h) $V_2$ for the reference image and images distorted with Gaussian white noise (GWN) and motion blur (MB).** Reprinted from [21] under a CC BY license, with permission from Eric Meier, original copyright [2007].

Where, $m \in 1, 2, \ldots, M$, $n \in 1, 2, \ldots, N$, while M and N represent the height and width of the image.

The histogram of the pairwise products of MSCN coefficients throughout the eight orientations is presented in Fig 5.

The difference between pairwise products of MSCN coefficients along $H_1$ and $H_2$, $V_1$ and $V_2$, $D_1$ and $D_3$, and $D_2$ and $D_4$ were calculated, which indicates that $H_1 = H_2$, $V_1 = V_2$, $D_1 = D_3$, and $D_2 = D_4$. Hence, four orientations, namely $H_1$, $V_1$, $D_1$ and $D_2$ were chosen for the AGGD calculations.

Four parameters were computed for AGGD, namely $\eta$, $v$, $\sigma_l^2$ and $\sigma_r^2$. Specifically, $v$ represents the shape of the distribution, $\sigma_l^2$ and $\sigma_r^2$ represent the left- and right-scale parameters, and $\eta$ represents the mean of the distribution. The four parameters of AGGD, namely $\eta$, $v$, $\sigma_l^2$, $\sigma_r^2$, were calculated using the formula in (16) [25]. The AGGD parameters, $\eta$, $v$, $\sigma_l^2$, $\sigma_r^2$ were calculated throughout $H_1$, $V_1$, $D_1$ and $D_2$ orientations as shown in Eq (17) and this forms 16 parameters of AGGD.

$$
f\left(x; v, \sigma_l^2, \sigma_r^2\right) = \begin{cases} \dfrac{v}{(\beta_l + \beta_r)\Gamma\left(\frac{1}{v}\right)} \exp\left(-\left(\dfrac{-x}{\beta_l}\right)^v\right) & x < 0 \\ \dfrac{v}{(\beta_l + \beta_r)\Gamma\left(\frac{1}{v}\right)} \exp\left(-\left(\dfrac{x}{\beta_r}\right)^v\right) & x \geq 0 \end{cases} \tag{16}
$$

Where:

$$
x = H_1(m, n), \ V_1(m, n), \ D_1(m, n), \ D_2(m, n) \tag{17}
$$

$$
\beta_l = \sigma_l \sqrt{\dfrac{\Gamma\left(\frac{1}{v}\right)}{\Gamma\left(\frac{3}{v}\right)}} \tag{18}
$$

$$
\beta_r = \sigma_r \sqrt{\dfrac{\Gamma\left(\frac{1}{v}\right)}{\Gamma\left(\frac{3}{v}\right)}} \tag{19}
$$

The parameters $(\eta,\ v, \sigma_l^2, \sigma_r^2)$ of the best AGGD fit were computed using the similar moment-matching approach, which was used for GGD, while $\eta$ was calculated using the formula in (20) [12]:

$$\eta = (\beta_r - \beta_l)\frac{\Gamma\left(\frac{2}{v}\right)}{\Gamma\left(\frac{1}{v}\right)} \tag{20}$$

In total, calculations were performed on 18 parameters of GGD and AGGD for the wood images, such as two parameters of GGD: $\alpha$, $\sigma^2$, 16 parameters of AGGD: four AGGD parameters, $\eta$, $v$, $\sigma_l^2$, $\sigma_r^2$ x 4 orientations, including $H_1$, $V_1$, $D_1$, and $D_2$, as shown in Table 1.

According to Mittal et al., accurate assessment of images could be conducted through IQA, which presents multi-scale information of an image [12]. Therefore, the aforementioned 18 parameters were computed at two scales (original image scale and image reduced by a factor of 0.5). Therefore, 36 parameters were generated from the full procedure to represent the features of wood images, and all parameters were used to train the SVR. As a result, only two scales were used, which reflected Mittal et al.'s statement that no improvement took place in the performance of the metric when more scales were incorporated [12]. The computation time would also increase with the increasing number of scales.

**MOS.** Ten students from the Department of Electrical and Electronics Engineering in Manipal International University (MIU), Nilai, Malaysia, who aged 20 to 25 years old, volunteered to evaluate the wood images. The evaluation was performed using a 21 inch LED monitor with a resolution of 1920 x 1080 pixels based on the procedures recommended in Rec. ITU-R BT.500-11 [26] within an office environment. The uncorrected near vision acuity of every subject was checked using the Snellen Chart prior to the subjective evaluation to confirm their fitness to perform the evaluation task.

After the examination of the uncorrected near vision acuity, a subjective evaluation was conducted. Consisting of a process which took 15 to 20 minutes, the evaluation was performed based on the Simultaneous Double Stimulus for Continuous Evaluation (SDSCE) methodology [26, 27]. In this case, the reference and distorted images were displayed on the monitor screen side-by-side, where the reference image was displayed on the left and the distorted image was displayed on the right. The distorted image was evaluated by each subject through the comparison between the distorted image (right side) and reference image (left side) in terms of quality. The image was either rated as Excellent (5), Good (4), Fair (3), Poor (2), or Bad (1) based on each displayed image. However, the numerical scores were not revealed to the subjects due to the potential bias created between the subjects [25]. The ratings obtained from the subjects were used to calculate MOS using the formula in (21) [28]:

$$MOS\left(p\right) = \frac{1}{N}\sum\nolimits_{i=1}^{N} S_{ip} \tag{21}$$

**Table 1. Explanation of 18 parameters.**

| Features | Description | Computation procedures |
|---|---|---|
| $f_1 - f_2$ | $\alpha$ and $\sigma^2$ | GGD features |
| $f_3 - f_6$ | $v$, $\eta$, $\sigma_l^2$ and $\sigma_r^2$ | Horizontal ($H_1$) AGGD features |
| $f_7 - f_{10}$ | $v$, $\eta$, $\sigma_l^2$ and $\sigma_r^2$ | Vertical ($V_1$) AGGD features |
| $f_{11} - f_{14}$ | $v$, $\eta$, $\sigma_l^2$ and $\sigma_r^2$ | Diagonal ($D_1$) AGGD features |
| $f_{15} - f_{18}$ | $v$, $\eta$, $\sigma_l^2$ and $\sigma_r^2$ | Diagonal ($D_2$) AGGD features |

Where $S_{ip}$ refers to the score by $i^{th}$ subject for $p^{th}$ image, while $N$ represents the number of human subjects as N = 10. The MOS values obtained for wood images were also used to train SVR.

**Regression module.** An epsilon-SVR, $\in$ − SVR model was used in this study [29]. As previously mentioned, the $\in$ − SVR was trained using MOS, 36 GGD, and AGGD features of wood images. Following the calculation of 36 image features for the wood images, mapping of the features to MOS values of the respective wood images were performed. The 36 features and MOS of wood images were then divided randomly into two sets, where one set was used for training and another set was used to test the system. While 80% of the 36 features and MOS values were used to train the SVR model, the remaining 20% were used to test the system. The training and testing datasets were permutated randomly to avoid any bias during the training and testing of the system [12].

The difference between BRISQUE and WNR-IQA could be seen from how BRISQUE is the generalised form of IQA, which is made to obtain quality score for natural images, while WNR-IQA is created specifically for wood images. Natural images are any natural light images which are captured by an optical camera without any pre-processing [12]. While, wood images are captured using a portable camera which has ten times magnification lens [3]. The differences between BRISQUE and WNR-IQA flowcharts are presented in Fig 6.

Pearson's Linear Correlation Coefficient (PLCC) [19] and Root Mean Square Error (RMSE) [20] between the MOS values and the quality score, which were obtained from the WNR-IQA, were calculated to evaluate the performance of the system. The accuracy of the system was indicated through higher PLCC and lower RMSE values due to the high similarity between the quality scores obtained from the WNR-IQA to the MOS values in terms of magnitude. The training and testing of the system were iterated 100 times, while the PLCC and RMSE values were recorded for every iteration. As a result, the medians of PLCC and RMSE

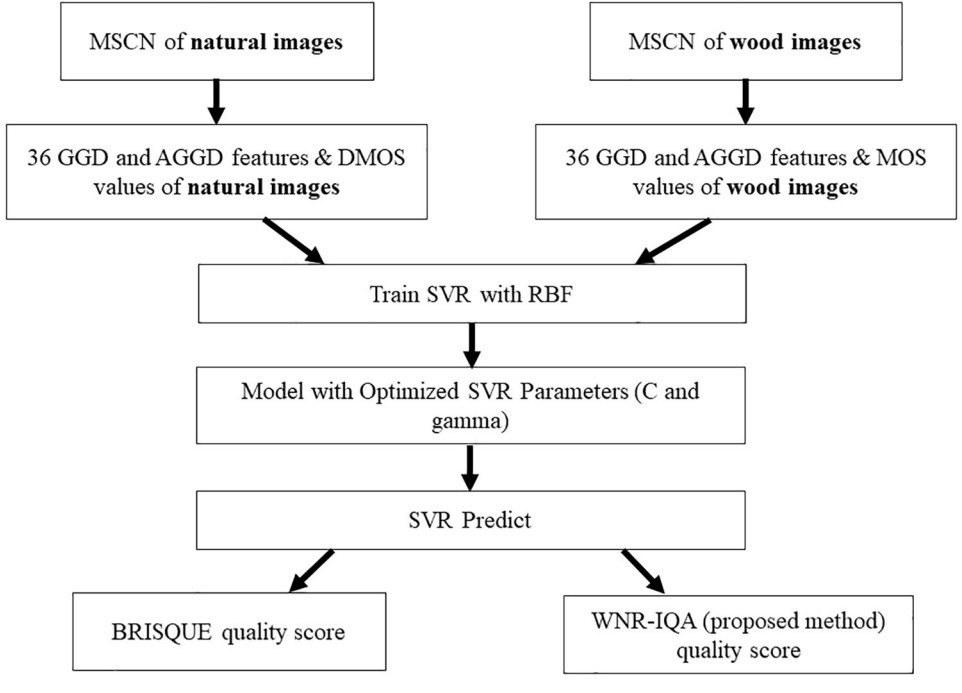

**Fig 6. Differences between BRISQUE and WNR-IQA.**

amounted to 0.935 and 0.361, respectively. Moreover, the optimised cost parameter (C) and width parameter (g) of the SVR model, which amounted to 512 and 0.25, respectively, were selected based on the median of the PLCC and RMSE values. Following that, these parameters were used to form the optimised SVR model.

## Performance evaluation

The second dataset was created specifically to evaluate the performance of WNR-IQA, where only the second dataset was used instead of the wood images highlighted in the Wood images sub-section for the evaluation of the performance of WNR-IQA. To illustrate, provided that the wood images were used for the training of the SVR, the second dataset was created to avoid any bias in performance evaluation. This dataset was generated using 10 'perfect' reference images obtained from ten different wood species, namely *Julbernardia pellegriniana* (Beli), *Dalbergia cultrate* (Blackwood), *Dalbergia retusa* (Cocobolo), *Dalbergia cearensis* (Kingwood), *Guaiacum officinale* (Lignum), *Swartzia spp.* (Queenwood), *Dalbergia spruceana* (Rosewood), *Dalbergia sissoo* (Sisso), *Swartzia benthamiana* (Wamara), and *Euxylophora paraensis* (Yellowheart). These images are presented in Fig 7.

These images were obtained from the same wood image database [21] and distorted with Gaussian white noise with $\sigma_{GN}$ = 10, 20, 30, 40, and 50, including a motion blur with $\sigma_{MB}$ = 2, 4, 6, 8, and 10. Using motion blur, further distortion was performed on the images distorted by the Gaussian white noise. In this case, following the distortion of images with $\sigma_{GN}$ = 10 was further distortion with $\sigma_{MB}$ = 2, 4, 6, 8 and 10, and these procedures were repeated for images distorted with $\sigma_{GN}$ = 20, 30, 40, and 50. Overall, 360 wood images were generated in the dataset.

The proposed WNR-IQA metric was compared with five FR-IQA metrics obtained for the second dataset, namely Structural Similarity Index (SSIM) [15], Multiscale SSIM (MS-SSIM) [15], Feature Similarity (FSIM) [16], Information Weighted SSIM (IW-SSIM) [17], and Gradient Magnitude Similarity Deviation (GMSD) [18]. In addition, the WNR-IQA was also

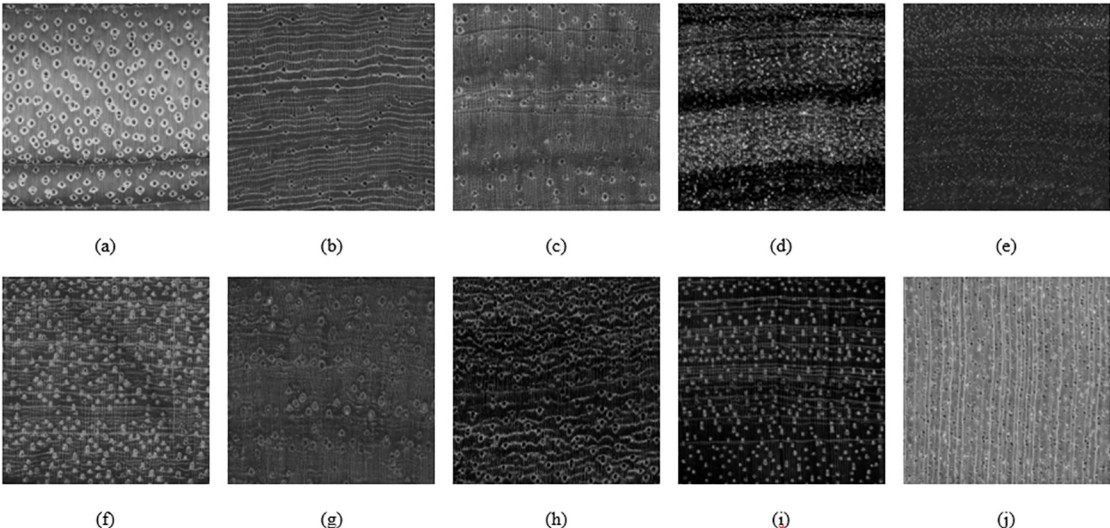

**Fig 7. Ten reference wood images in the second dataset (a)** *Julbernardia pellegriniana,* **(b)** *Dalbergia cultrate,* **(c)** *Dalbergia retusa,* **(d)** *Dalbergia cearensis,* **(e)** *Guaiacum officinale,* **(f)** *Swartzia spp.,* **(g)** *Dalbergia spruceana,* **(h)** *Dalbergia sissoo,* **(i)** *Swartzia benthamiana,* **and (j)** *Euxylophora paraensis.* Reprinted from [21] under a CC BY license, with permission from Eric Meier, original copyright [2007].

compared with BRISQUE [13], deepIQA [12], and DB-CNN [14] obtained for the second dataset. This was followed by the calculation of PLCC and RMSE [19] values between the FR-I-QAs, BRISQUE, deepIQA, DB-CNN, and WNR-IQA for the evaluation of the performance of the WNR-IQA, BRISQUE, deepIQA, DB-CNN, and FR-IQAs.

## Results and discussions

### The relationship between MOS and different distortion levels

The relationship between MOS and different distortion levels of Gaussian white noise, motion blur, and a mixture of Gaussian white noise and motion blur is presented from Fig 8a–8g. Higher MOS values indicated higher image quality, while higher distortion levels represented lower image quality. Therefore, lower MOS values would be produced for images with higher distortion levels. Based on the scatter plot presented in Fig 8a–8e, the MOS value was reduced with the increase in distortion level. Accordingly, it was indicated that human subjects were able to differentiate the images distorted with different levels of Gaussian white noise, motion blur, and the mixture of both distortions. It could be seen from the scatter plot in Fig 8f and 8g that the MOS value amounted to 1, while the images distorted with Gaussian white noise, $\sigma_{GN}$, amounted to 40 and 50 at all the levels of motion blur due to the poor quality of the images.

### Relationship between MOS and proposed WNR-IQA, BRISQUE, deepIQA, DB-CNN, FR-IQAs

The calculated PLCC and RMSE values between MOS and the WNR-IQA, BRISQUE, deepIQA, DB-CNN, and the five FR-IQA metrics are presented in Table 2. PLCC values close to 1, indicate a close correlation of MOS with the IQA metric, while lower RMSE values indicate a correlation of MOS with the IQA metric. Table 2 shows that the highest PLCC values were recorded for Gaussian white noise, motion blur, the mixture of Gaussian white noise and motion blur, and the overall database obtained for the WNR-IQA compared to BRISQUE,

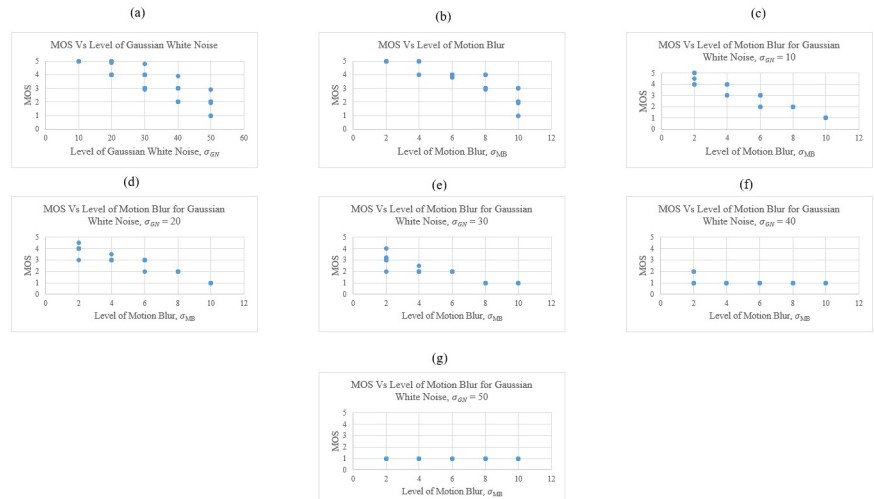

**Fig 8. Scatter plot of MOS versus distortion levels of (a) Gaussian white noise, (b) Motion blur, (c) Motion blur for Gaussian white noise, $\sigma_{GN}$ = 10, (d) Motion blur for Gaussian white noise, $\sigma_{GN}$ = 20, (e) Motion blur for Gaussian white noise, $\sigma_{GN}$ = 30, (f) Motion blur for Gaussian white noise, $\sigma_{GN}$ = 40 and (g) Motion blur for Gaussian white noise, $\sigma_{GN}$ = 50.** Reprinted from [21] under a CC BY license, with permission from Eric Meier, original copyright [2007].

**Table 2. PLCC and RMSE values between MOS and WNR-IQA, BRISQUE, deepIQA, DB-CNN, and FR-IQAs.**

|      |                      | WNR-IQA | BRISQUE | deepIQA | DB-CNN | MSSIM | SSIM  | FSIM  | IWSSIM | GMSD  |
|------|----------------------|---------|---------|---------|--------|-------|-------|-------|--------|-------|
| PLCC | GWN                  | **0.928** | 0.592 | 0.675 | 0.759 | 0.857 | 0.871 | 0.898 | 0.865 | 0.894 |
|      | MB                   | **0.960** | 0.586 | 0.698 | 0.725 | 0.863 | 0.818 | 0.920 | 0.914 | 0.865 |
|      | Mixture of GWN and MB | **0.925** | 0.591 | 0.642 | 0.698 | 0.861 | 0.832 | 0.895 | 0.880 | 0.872 |
|      | All                  | **0.943** | 0.612 | 0.639 | 0.727 | 0.850 | 0.815 | 0.903 | 0.876 | 0.895 |
| RMSE | GWN                  | **0.481** | 1.039 | 0.945 | 0.867 | 0.664 | 0.633 | 0.568 | 0.647 | 0.578 |
|      | MB                   | **0.321** | 0.925 | 0.923 | 0.910 | 0.577 | 0.656 | 0.448 | 0.462 | 0.572 |
|      | Mixture of GWN and MB | **0.352** | 0.930 | 0.929 | 0.873 | 0.624 | 0.645 | 0.529 | 0.551 | 0.574 |
|      | All                  | **0.385** | 0.916 | 0.911 | 0.890 | 0.623 | 0.721 | 0.456 | 0.529 | 0.539 |

deepIQA, DB-CNN, and five FR-IQAs. Therefore, WNR-IQA displayed a higher performance compared to BRISQUE, deepIQA, DB-CNN, SSIM, MS-SSIM, FSIM, IW-SSIM, and GMSD.

It is also indicated from Table 2 that the lowest PLCC values were recorded for BRISQUE, indicating that BRISQUE was not compatible with the assessment of wood images. This incompatibility was also indicated by the highest RMSE values recorded for BRISQUE. However, WNR-IQA had a higher performance compared to BRISQUE, deepIQA, DB-CNN, and FR-IQAs as it was adapted for wood images. The model was also trained with GGD and AGGD features, including the MOS obtained for wood images unlike BRISQUE, deepIQA, DB-CNN, and FR-IQAs, which were designed based on the features and their similarities, luminance, contrast, and structure of natural images. Additionally, WNR-IQA also had a higher performance compared to FR-IQAs as it does not require a perfect reference image.

## Conclusion

In this article, Wood No-Reference Image Quality Assessment (WNR-IQA), was proposed for the evaluation of wood images prior to classification of species. Provided that the established NR-IQA metrics, BRISQUE, deepIQA and DB-CNN were designed for the assessment of natural images, they were not optimal for the assessment of wood images. Therefore, the WNR-IQA was trained using MOS and a set of features calculated specifically for wood images. This was followed by the evaluation of the performance of the WNR-IQA by comparing the correlation between MOS, WNR-IQA, BRISQUE, deepIQA, DB-CNN, and five FR-IQA metrics using PLCC and RMSE. It was indicated from the values of PLCC and RMSE that WNR-IQA exhibited higher performance compared to BRISQUE, deepIQA, DB-CNN, and the five FR-IQAs. Furthermore, the proposed WNR-IQA performed an accurate assessment of the quality of wood images, which should function in the selection of suitable images to be included in the wood recognition algorithm. Essentially, the acquirement of a perfect image is impossible in the timber industry due to its environment, which consists of dust, poor illumination, hot environment, and motion blur caused by relative motion between the camera and the wood slice. However, the quality assessment in this study did not require a perfect reference image for the evaluation of the quality of the test wood images.

## Acknowledgments

The authors would like to thank Eric Meier, the creator of The Wood Database, for providing the wood samples, and Professor Paul Cumming for performing a critical reading of the manuscript.

## Author Contributions

**Conceptualization:** Heshalini Rajagopal.

**Data curation:** Heshalini Rajagopal.

**Formal analysis:** Heshalini Rajagopal.

**Funding acquisition:** Heshalini Rajagopal.

**Investigation:** Heshalini Rajagopal.

**Methodology:** Heshalini Rajagopal.

**Project administration:** Heshalini Rajagopal.

**Resources:** Heshalini Rajagopal.

**Software:** Heshalini Rajagopal.

**Supervision:** Norrima Mokhtar.

**Visualization:** Heshalini Rajagopal.

**Writing – original draft:** Heshalini Rajagopal, Norrima Mokhtar, Tengku Faiz Tengku Mohmed Noor Izam, Wan Khairunizam Wan Ahmad.

**Writing – review & editing:** Heshalini Rajagopal, Norrima Mokhtar, Tengku Faiz Tengku Mohmed Noor Izam, Wan Khairunizam Wan Ahmad.

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
