## [Decision Letter · Decision Letter 0]

20 Jan 2020

PONE-D-19-34649

No-Reference Quality Assessment for Image-Based Assessment of Economically Important Tropical Woods

PLOS ONE

Dear Dr Mokhtar,

Thank you for submitting your manuscript to PLOS ONE. After careful consideration, we feel that it has merit but does not fully meet PLOS ONE’s publication criteria as it currently stands. Therefore, we invite you to submit a revised version of the manuscript that addresses the points raised during the review process.

We would appreciate receiving your revised manuscript by Mar 05 2020 11:59PM. To enhance the reproducibility of your results, we recommend that if applicable you deposit your laboratory protocols in protocols.io, where a protocol can be assigned its own identifier (DOI) such that it can be cited independently in the future. For instructions see: http://journals.plos.org/plosone/s/submission-guidelines#loc-laboratory-protocols

We look forward to receiving your revised manuscript.

Kind regards,

Yiming Tang, Ph.D.

Academic Editor

PLOS ONE

Journal Requirements:

2. Please amend your Data availability statement to provide links/URLs to how other researchers may access the images and data used in this study.

4. We note that Figures 1-8 in your submission contain copyrighted images. All PLOS content is published under the Creative Commons Attribution License (CC BY 4.0), which means that the manuscript, images, and Supporting Information files will be freely available online, and any third party is permitted to access, download, copy, distribute, and use these materials in any way, even commercially, with proper attribution. For more information, see our copyright guidelines: http://journals.plos.org/plosone/s/licenses-and-copyright.

1.         You may seek permission from the original copyright holder of Figures 1-8 to publish the content specifically under the CC BY 4.0 license. 

Additional Editor Comments:

The work was not convincing in the current version. Two reviewers pointed out some tough comments, which required further in-depth study and response by the authors. It was demanded to provide the comparison with related top algorithms (mentioned by Reviewer 2).

Reviewers' comments:

Reviewer's Responses to Questions

**Comments to the Author**

1. Is the manuscript technically sound, and do the data support the conclusions?

Reviewer #1: Partly

Reviewer #2: No

2. Has the statistical analysis been performed appropriately and rigorously? 

Reviewer #1: No

Reviewer #2: No

3. Have the authors made all data underlying the findings in their manuscript fully available?

Reviewer #1: Yes

Reviewer #2: Yes

4. Is the manuscript presented in an intelligible fashion and written in standard English?

Reviewer #1: No

Reviewer #2: Yes

5. Review Comments to the Author

Reviewer #1: In this paper, the authors proposed a No-Reference IQA (NR-IQA) metric to assess the quality of wood images, Wood NR-IQA (WNR-IQA). Support Vector Machine (SVM) Regression (SVR) was trained using Generalized Gaussian Distribution (GGD) and Asymmetric Generalized Gaussian Distribution (AGGD) features calculated for wood images and the mean opinion score (MOS) obtained from subjective evaluation. Compared to other six metric, namely Blind/Referenceless Image Spatial Quality Evaluator (BRISQUE) and five Full Reference-IQA (FR-IQA) metrics known as MSSIM, SSIM, FSIM, IWSSIM and GMSD, the proposed WNR-IQA had better performance in PLCC and RMSE values. Furthermore, it is meaningful that the proposed metric does not require a “perfect” reference image in order to evaluate the images.

But, this manuscript in this current form has some problems as follows:

1) The introduction is not enough because the references for the current researches are too few while the related analyses are not profound.

2) The metric most relevant to the proposed metric is the BRISQUE. But the authors did not show the differences between them, especially the difference in image features adoption and the difference in procedure of these two metrics. Please add relevant instructions to highlight the innovation of this paper.

3) In Eq. (4) and (7), what does the “x” mean?

4) In section “Results and Discussions”, the authors have only considered the Gaussian white noise and motion blur. Why the mixture of these two noises were ignored in simulating the image distortions encountered in the industrial setting?

5) In section “Results and Discussions”, why the ten wood images mentioned in “wood images” , namely Avodire, Bocote, Butternut, Basswood, Dogwood, Laurel Blanco, Katalox, and Keruing are not taken in to experiments? They are replaced by other ten species, why?

6) In section “Relationship between MOS and Proposed WNR-IQA, BRISQUE,FR-IQAs”, there is no detailed explanation about Fig 8. In addition, pictures shown in Fig 8a-u are blurred. It is unacceptable.

7) Please clearly explain the reasons in section “Results and Discussions” based on the expremental results, why the proposed metric WNR-IQA is superior to other six metrics? And why other related metrics have lower performance?

8) The 18 parameters of GGD and AGGD for wood images are keys for proposed WNR-IQA. But there are no equations or detailed instructions to show how to calculate these parameters in this paper. That should not be neglected or omitted.

9) English expressions need to be carefully checked out. For example, in the first paragraph of Introduction, there exist the following sentence (noting “it high strength and density”):

“Rosewood (Dalbergia sp.) is another expensive wood, sought after for instrument making and flooring due to it high strength and density.”

Besides, at line 111 to 112, there is a sentence (noting “were the distorted”):

“These ten reference wood images were the distorted by Gaussian white noise and motion blur, which represent image distortions typically encountered in the industrial setting.”

Totally speaking, this manuscript is not well-prepared.

Reviewer #2: The authors presented a No-Reference IQA metric to assess the quality of wood images. Some experiments are done and compared with some other metrics. The results are rather OK. However, I can not find the innovative points in the paper since the authors just utilize the regression module to assess the quality of wood images. Furthermore, many state-of-the-art references are not cited. The experiments just compared with traditional methods, and many recent methods, especial some deep learning based methods such as Hallucinated-IQA and deepIQA, are not used to compare.

Minor points:

What’s mean of \\in -SVR in Line 239?

In equ. 19, it’s better to add subscript k in MOS since you calculate the MOS of the k^{th} image.

6. PLOS authors have the option to publish the peer review history of their article (what does this mean?). If published, this will include your full peer review and any attached files.

Reviewer #1: No

Reviewer #2: No

---

## [Author Response · Author response to Decision Letter 0]

4 Mar 2020

5 March 2020

Dear Editor and Reviewers,

Thank you very much for considering our manuscript entitled “No-reference quality assessment for image-based assessment of economically important tropical woods”. We really appreciate your valuable suggestions and comments which help us to enhance the quality of our paper. We have revised the manuscript accordingly, with details below:

Reviewers' comments:

Please ensure that your manuscript meets PLOS ONE's style requirements, including those for file naming. The PLOS ONE style templates can be found at http://www.journals.plos.org/plosone/s/file?id=wjVg/PLOSOne_formatting_sample_main_body.pdf and 

http://www.journals.plos.org/plosone/s/file?id=ba62/PLOSOne_formatting_sample_title_authors_affiliations.pdf

 The manuscript has been revised according to the PLOS ONE's style.

 Please amend your Data availability statement to provide links/URLs to how other researchers may access the images and data used in this study.

 The images and data used in this study are publicly available in https://github.com/Heshalini/Wood-Image-Quality/tree/b03346da9d1efb887ec6d390f235822cc9470f49. 

 We suggest you thoroughly copyedit your manuscript for language usage, spelling, and grammar. If you do not know anyone who can help you do this, you may wish to consider employing a professional scientific editing service. 

 The manuscript has been proofread and edited by professional proofreading and editing service, Proofreading by A UK PhD.

 We note that Figures 1-8 in your submission contain copyrighted images. All PLOS content is published under the Creative Commons Attribution License (CC BY 4.0), which means that the manuscript, images, and Supporting Information files will be freely available online, and any third party is permitted to access, download, copy, distribute, and use these materials in any way, even commercially, with proper attribution.

 Figs 1-8 were generated using the publicly available wood images database owned by Mr Eric Meier. Permission to publish the figures has been obtained from Mr Eric Meier and the completed Content Permission Form and e-mail from Mr Eric which mentioned that he permits the usage of the wood images from his website are uploaded as an "Other" file with this submission. 

 Reviewer 1:

In this paper, the authors proposed a no-reference IQA (NR-IQA) metric to assess the quality of wood images, Wood NR-IQA (WNR-IQA). Support Vector Machine (SVM) Regression (SVR) was trained using Generalized Gaussian Distribution (GGD) and Asymmetric Generalized Gaussian Distribution (AGGD) features calculated for wood images and the mean opinion score (MOS) obtained from the subjective evaluation. Compared to other six metrics, namely Blind/Referenceless Image Spatial Quality Evaluator (BRISQUE) and five Full Reference-IQA (FR-IQA) metrics known as MSSIM, SSIM, FSIM, IWSSIM and GMSD, the proposed WNR-IQA had better performance in PLCC and RMSE values. Furthermore, it is meaningful that the proposed metric does not require a “perfect” reference image in order to evaluate the images.

But, this manuscript in this current form has some problems as follows:

 The introduction is not enough because the references for the current researches are too few while the related analyses are not profound.

 Explanation on current researches done on CNN based NR-IQAs, deepIQA [12] and Deep Bilinear Convolutional Neural Networks (DB-CNN) [14] were added in line 81-106 of the Introduction section. DeepIQA and DB-CNN are CNN based NR-IQAs where deepIQA is trained end-to-end and involves 10 convolutional layers, 5 pooling layers for feature extraction and 2 fully connected layers for regression [12] while DB-CNN is trained by two sets of features namely, CNN for synthetic distortions (S-CNN) and VGG-16, that are bi-linearly pooled to predict the quality of the image [14]. However, CNN based NR-IQA model requires a very large training database as a limited number of labelled training data often leads to overfitting problem in CNN [14].

 The metric most relevant to the proposed metric is the BRISQUE. But the authors did not show the differences between them, especially the difference in image features adoption and the difference in the procedure of these two metrics. Please add relevant instructions to highlight the innovation of this paper.

 The innovation of the proposed WNR-IQA model is that it is a No Reference – Image Quality Assessment (NR-IQA) model designed specifically for wood images. WNR-IQA is motivated by NR-IQA model which is independent of reference natural images. In fact, for practicality, good reference images especially given the nature of the wood industry (dusty environment and poor illumination) are not easily available. These were written in line 78-80 of the Introduction section.

 The proposed model uses a similar concept designed for natural images, with mathematical models that consider several factors like contrast, luminance, image feature, and Natural Scene Statistics (NSS). The key difference between the BRISQUE and WNR-IQA is that the BRISQUE was developed by training the SVR using the image features computed for natural images; whereas the WNR-IQA used the wood images for this calculation. These were written in line 277-282 of the Regression module section. Their differences were also illustrated in Fig 6, Regression module section.

 In Eq. (4) and (7), what does the “x” mean?

 Eq. (4) is the equation to calculate GGD features. x in Eq. (4) is the Mean Subtracted Contrast Normalized (MSCN), I ^(m,n) of wood images. This was written in Eq. (5), The features of GGD and AGGD section.

 Eq. (7) (in previous manuscript version) is now Eq. (16) which is the equation to calculate AGGD features. x in Eq. (16) is the pairwise products of MSCN coefficients which are computed along four orientation, H_1,V_1,D_1 and D_2. This was written in Eq. (17), The features of GGD and AGGD section 

 In section “Results and Discussions”, the authors have only considered the Gaussian white noise and motion blur. Why the mixture of these two noises were ignored in simulating the image distortions encountered in the industrial setting?

 All the ten wood images in the second dataset have been distorted with the mixture of Gaussian white noise and motion blur. The images which were distorted by Gaussian white noise were further distorted with motion blur, i.e. images distorted with σ_GN = 10 were further distorted with σ_MB = 2, 4, 6, 8 and 10 and the same procedure were repeated for images distorted with σ_GN = 20, 30, 40 and 50. In total, this dataset comprises of 360 wood images. These were written in line 314 – 317 of the Performance evaluation section. The results obtained for the mixture of Gaussian white noise and motion blur can be found in line 327– 365 of the Results and discussions section. 

 In section “Results and Discussions”, why the ten wood images mentioned in “wood images”, namely Avodire, Bocote, Butternut, Basswood, Dogwood, Laurel Blanco, Katalox, and Keruing are not taken into experiments? They are replaced by other ten species, why?

 The ten wood images mentioned in the wood images section are used to train and test the SVR model. These images were distorted with Gaussian White Noise and Motion Blur and were used to train and test the SVR model for 100 times where 100 PLCC and RMSE values were obtained and the median of these values were chosen in order to select the optimized cost parameter, C, and width parameter, g, of the SVR model. These are written in line 269-293 of the Regression module section. The median values of PLCC and RMSE were also mentioned in these lines.

 The performance of the proposed WNR-IQA metric is evaluated using the second dataset which also comprises of different ten wood images. Therefore, the results obtained for these images were included and discussed in the Results and Discussion section. The second dataset is used to evaluate the performance of WNR-IQA metric instead of the ten wood images mentioned in the wood images section (used to train SVR) to avoid any bias. These are written in line 295-299 of the Performance evaluation section.

 In section “Relationship between MOS and Proposed WNR-IQA, BRISQUE, FR-IQAs”, there is no detailed explanation about Fig 8. In addition, pictures shown in Fig 8a-u are blurred. It is unacceptable.

 Fig 8a-u which shows the distribution of the WNR-IQA, BRISQUE, FR-IQAs has been removed due to the poor quality of the images. Moreover, the relationship between MOS and Proposed WNR-IQA, BRISQUE, FR-IQAs is shown through the PLCC and RMSE values in Table 2, Relationship between MOS and proposed WNR-IQA, BRISQUE, deepIQA, DB-CNN, FR-IQAs section.

 Please clearly explain the reasons in section “Results and Discussions” based on the experimental results, why the proposed metric WNR-IQA is superior to the other six metrics? And why other related metrics have lower performance?

 The WNR-IQA was compared with BRISQUE, deepIQA, DB-CNN, five FR-IQAs: SSIM, MS-SSIM, FSIM, IW-SSIM and GMSD obtained for the second dataset. WNR-IQA outperforms BRISQUE, deepIQA, DB-CNN and FR-IQAs as it is tailored for wood images where the model has been trained with GGD and AGGD features and MOS obtained for wood images specifically unlike BRISQUE, deepIQA, DB-CNN and FR-IQAs which were designed by considering the features and similarities of features, luminance, contrast and structure of natural images. Furthermore, WNR-IQA is better than the FR-IQAs as it does not require a perfect reference image. These were written in line 347 – 365 of the Relationship between MOS and proposed WNR-IQA, BRISQUE, deepIQA, DB-CNN, FR-IQAs section.

 The 18 parameters of GGD and AGGD for wood images are keys for proposed WNR-IQA. But there are no equations or detailed instructions to show how to calculate these parameters in this paper. That should not be neglected or omitted.

 The equations and detailed instruction for GGD and AGGD parameters were added and can be found in line 182-234 and Table 1 of The features of GGD and AGGD section.

 English expressions need to be carefully checked out. For example, in the first paragraph of Introduction, there exist the following sentence (noting “it high strength and density”):

“Rosewood (Dalbergia sp.) is another expensive wood, sought after for instrument making and flooring due to its high strength and density.”

Besides, at line 111 to 112, there is a sentence (noting “were the distorted”):

“These ten reference wood images were distorted by Gaussian white noise and motion blur, which represent image distortions typically encountered in the industrial setting.”

 The above sentences were corrected and the manuscript has been proofread and edited by professional proofreading and editing service, Proofreading by A UK PhD.

 Reviewer #2: The authors presented a no-reference IQA metric to assess the quality of wood images. Some experiments are done and compared with some other metrics. The results are rather OK. However, I can not find the innovative points in the paper since the authors just utilize the regression module to assess the quality of wood images. Furthermore, many state-of-the-art references are not cited. The experiments just compared with traditional methods, and many recent methods, especial some deep learning-based methods such as Hallucinated-IQA and deepIQA, are not used to compare.

 The innovation of the proposed WNR-IQA model is that it is a No Reference – Image Quality Assessment (NR-IQA) model designed specifically for wood images. WNR-IQA is motivated by NR-IQA model which is independent of reference natural images. In fact, for practicality, good reference images especially given the nature of the wood industry (dusty environment and poor illumination) are not easily available. These were written in line 78 - 80 of the Introduction section.

 The proposed model uses a similar concept designed for natural images, with mathematical models that consider several factors like contrast, luminance, image feature, and Natural Scene Statistics (NSS). The key difference between the BRISQUE and WNR-IQA is that the BRISQUE was developed by training the SVR using the image features computed for natural images; whereas the WNR-IQA used the wood images for this calculation. These were written in the line 277 - 282 of the Regression module section. Their differences were also illustrated in Fig 6.

 The WNR-IQA was also compared with deepIQA and DB-CNN obtained for the second dataset. Results shows that WNR-IQA outperforms these two NR-IQAs as well. These were written in line 347 – 365 of the Relationship between MOS and proposed WNR-IQA, BRISQUE, deepIQA, DB-CNN, FR-IQAs section.

Minor points:

What’s mean of \\in -SVR in Line 239?

 ∈-SVR is epsilon-SVR which is a type of SVR model. This was written in line 269 of the Regression module section.

In equ. 19, it’s better to add subscript k in MOS since you calculate the MOS of the k^{th} image.

 Subscript k was replaced with p to avoid confusion with the window sizes in Eq. (2) and (3). Subscript p was added in MOS equation in Eq. (21), MOS section. 

Thank you for your kind consideration of our work. 

Yours Faithfully,

NORRIMA MOKHTAR

Senior Lecturer

Department of Electrical Engineering

Faculty of Engineering

University of Malaya

50603 Kuala Lumpur

Malaysia

---

## [Decision Letter · Decision Letter 1]

22 Apr 2020

PONE-D-19-34649R1

No-Reference Quality Assessment for Image-Based Assessment of Economically Important Tropical Woods

PLOS ONE

Dear Dr Mokhtar,

Thank you for submitting your manuscript to PLOS ONE. After careful consideration, we feel that it has merit but does not fully meet PLOS ONE’s publication criteria as it currently stands. Therefore, we invite you to submit a revised version of the manuscript that addresses the points raised during the review process.

We would appreciate receiving your revised manuscript by Jun 06 2020 11:59PM. To enhance the reproducibility of your results, we recommend that if applicable you deposit your laboratory protocols in protocols.io, where a protocol can be assigned its own identifier (DOI) such that it can be cited independently in the future. For instructions see: http://journals.plos.org/plosone/s/submission-guidelines#loc-laboratory-protocols

We look forward to receiving your revised manuscript.

Kind regards,

Yiming Tang, Ph.D.

Academic Editor

PLOS ONE

Reviewers' comments:

Reviewer's Responses to Questions

**Comments to the Author**

1. If the authors have adequately addressed your comments raised in a previous round of review and you feel that this manuscript is now acceptable for publication, you may indicate that here to bypass the “Comments to the Author” section, enter your conflict of interest statement in the “Confidential to Editor” section, and submit your "Accept" recommendation.

Reviewer #1: All comments have been addressed

Reviewer #2: All comments have been addressed

2. Is the manuscript technically sound, and do the data support the conclusions?

Reviewer #1: Yes

Reviewer #2: Yes

3. Has the statistical analysis been performed appropriately and rigorously? 

Reviewer #1: Yes

Reviewer #2: Yes

4. Have the authors made all data underlying the findings in their manuscript fully available?

Reviewer #1: Yes

Reviewer #2: Yes

5. Is the manuscript presented in an intelligible fashion and written in standard English?

Reviewer #1: Yes

Reviewer #2: Yes

6. Review Comments to the Author

Reviewer #1: The authors have followed carefully our suggestions, and therefore I recommend accepting this paper for publication in PLOS ONE.

Reviewer #2: My comments are responded by authors carefully. The manuscript is improvement significantly. However, the model description is still not clear enough. Please give some explanation the difference between "natural image" and "wood image" . In Line 216, a parameter \\eta is missed

7. PLOS authors have the option to publish the peer review history of their article (what does this mean?). If published, this will include your full peer review and any attached files.

Reviewer #1: No

Reviewer #2: No

---

## [Author Response · Author response to Decision Letter 1]

28 Apr 2020

29 April 2020

Dear Editor and Reviewers,

Thank you very much for considering our manuscript entitled “No-reference quality assessment for image-based assessment of economically important tropical woods”. We really appreciate your valuable suggestions and comments which help us to enhance the quality of our paper. We have revised the manuscript accordingly, with details below:

Reviewers' comments:

 Reviewer #2: My comments are responded by authors carefully. The manuscript is improvement significantly. However, the model description is still not clear enough. Please give some explanation the difference between "natural image" and "wood image" . In Line 216, a parameter \\eta is missed

 The features of GGD and AGGD section was revised to make the model description clear. The content of the features of GGD and AGGD section was arranged to be in a flow to make the model description clear. Furthermore, line 227- 228 was revised by adding the following statement: 

“The AGGD parameters, η,v,σ_l^2,σ_r^2 were calculated throughout H_1, V_1, D_1 and D_2 orientations as shown in equation (17) and this forms 16 parameters of AGGD.” 

Table 1 which explains the 18 GGD and AGGD parameters was corrected as there was an error previously. The revised content can be found in line 146 – 250.

Fig 6 which shows the differences between BRISQUE and WNR-IQA was revised where the MSCN for the images was added into the flowchart to make the flowchart of the model clear. 

 The natural and wood images were explained in line 283-285, Regression module section. Natural images are any natural light images which are captured by an optical camera without any pre-processing. While, wood images are captured using a portable camera which has ten times magnification lens.

 The missing parameter \\eta in line 216 has been revised. The revised line can be found in line 224, The features of GGD and AGGD section. 

Thank you for your kind consideration of our work. 

Yours Faithfully,

NORRIMA MOKHTAR

Senior Lecturer

Department of Electrical Engineering

Faculty of Engineering

University of Malaya

50603 Kuala Lumpur

Malaysia

---

## [Decision Letter · Decision Letter 2]

4 May 2020

No-reference quality assessment for image-based assessment of economically important tropical woods

PONE-D-19-34649R2

Dear Dr. Mokhtar,

We are pleased to inform you that your manuscript has been judged scientifically suitable for publication and will be formally accepted for publication once it complies with all outstanding technical requirements.

With kind regards,

Yiming Tang, Ph.D.

Academic Editor

PLOS ONE

Reviewers' comments:

Reviewer's Responses to Questions

**Comments to the Author**

1. If the authors have adequately addressed your comments raised in a previous round of review and you feel that this manuscript is now acceptable for publication, you may indicate that here to bypass the “Comments to the Author” section, enter your conflict of interest statement in the “Confidential to Editor” section, and submit your "Accept" recommendation.

Reviewer #1: All comments have been addressed

Reviewer #2: All comments have been addressed

2. Is the manuscript technically sound, and do the data support the conclusions?

Reviewer #1: Yes

Reviewer #2: Yes

3. Has the statistical analysis been performed appropriately and rigorously? 

Reviewer #1: Yes

Reviewer #2: Yes

4. Have the authors made all data underlying the findings in their manuscript fully available?

Reviewer #1: Yes

Reviewer #2: Yes

5. Is the manuscript presented in an intelligible fashion and written in standard English?

Reviewer #1: Yes

Reviewer #2: Yes

6. Review Comments to the Author

Reviewer #1: The authors have followed carefully our suggestions and I recommend accepting this paper for publication in PLOS ONE.

Reviewer #2: All my comments are considered. I have no any further suggestion. The manuscript can be accepted now.

7. PLOS authors have the option to publish the peer review history of their article (what does this mean?). If published, this will include your full peer review and any attached files.

Reviewer #1: No

Reviewer #2: Yes: Duanbing Chen

---

## [Editor Report · Acceptance letter]

7 May 2020

PONE-D-19-34649R2 

No-reference quality assessment for image-based assessment of economically important tropical woods 

Dear Dr. Mokhtar:

I am pleased to inform you that your manuscript has been deemed suitable for publication in PLOS ONE. Congratulations! Your manuscript is now with our production department. 

With kind regards,

on behalf of

Professor Yiming Tang 

Academic Editor

PLOS ONE